# Computer-Based Intelligent Solutions for the Diagnosis of Gastroesophageal Reflux Disease Phenotypes and Chicago Classification 3.0

**DOI:** 10.3390/healthcare11121790

**Published:** 2023-06-17

**Authors:** Yunus Doğan, Serhat Bor

**Affiliations:** 1Department of Computer Engineering, Dokuz Eylül University, Izmir 35390, Türkiye; 2Department of Gastroenterology, Ege University Faculty of Medicine, Bornova, Izmir 35100, Türkiye; serhat.bor@ege.edu.tr

**Keywords:** artificial intelligence, healthcare systems, phenotyping

## Abstract

Gastroesophageal reflux disease (GERD) is a multidisciplinary disease; therefore, when treating GERD, a large amount of data needs to be monitored and managed.The aim of our study was to develop a novel automation and decision support system for GERD, primarily to automatically determine GERD and its Chicago Classification 3.0 (CC 3.0) phenotypes. However, phenotyping is prone to errors and is not a strategy widely known by physicians, yet it is very important in patient treatment. In our study, the GERD phenotype algorithm was tested on a dataset with 2052 patients and the CC 3.0 algorithm was tested on a dataset with 133 patients. Based on these two algorithms, a system was developed with an artificial intelligence model for distinguishing four phenotypes per patient. When a physician makes a wrong phenotyping decision, the system warns them and provides the correct phenotype. An accuracy of 100% was obtained for both GERD phenotyping and CC 3.0 in these tests. Finally, since the transition to using this developed system in 2017, the annual number of cured patients, around 400 before, has increased to 800. Automatic phenotyping provides convenience in patient care, diagnosis, and treatment management. Thus, the developed system can substantially improve the performance of physicians.

## 1. Introduction

Similar to studies in other departments of medicine, in studies of gastroesophageal reflux disease (GERD), data size is very important in obtaining accurate and reliable analysis results. However, recent studies described in the literature have been conducted with very little data. For example, 114 patients were evaluated in an ulcerative colitis study [1], 122 patients were analyzed in an inflammatory bowel disease study [2], and 400 patients were examined in a gastric cancer study [3]. A sample size of 400 patients is too low for accurate and reliable study of a disease as prevalent as gastric cancer. However, nowadays, large amounts of data can be stored within a regular-sized structure using a central database. Moreover, this type of information system can store patient data with distinctive characteristic features, such as different histories, sociodemographic data, etc. [4,5]. Thus, diverse data from various patient profiles have been used to determine the rules needed to create a decision support system (DSS) [6]. Nowadays, computer-based intelligent solutions are a necessity, and their use is widespread, including in the medical sector. Recent medical studies in the fields of data mining [7], artificial intelligence (AI) [8], machine learning, and deep learning have been conducted in relation to subjects such as medical image processing using radiological data [9] and early diagnosis of the deadliest diseases, such as heart disease, cancer, and diabetes [10,11]. In these studies, more than 85,000 patients have been analyzed to uncover more about these diseases, and these data have been used to develop information systems. The Ege University Medical School, Division of Gastroenterology, Reflux Center, which has the largest number of patients in Turkiye, carried out a scientific research project (2015-TIP-070) on switching from a Microsoft-Access-based system, which was limited to analysis, to a web application (using ASP.NET technology) with a database (Microsoft SQL) with advanced reporting functions. This developed system is now the biggest database in Turkiye in terms of the number of patients, with data recorded from more than 8000 patients from 2017 to 2020. Although patient care and health recording were disrupted during the COVID-19 pandemic, information on a total of 12,000 patients was included by 2022. Many new algorithms can be created using data mining techniques with such a large patient series. In addition, intelligent software can be used to detect false-positive and false-negative rates in this comprehensive database of patient information. Furthermore, as the GERD phenotypes are not widely known among physicians, they are either not recognized or incorrectly identified. Thus, the created database and smart learning system was designed to help physicians in this sense. As a result, the system aims for GERD patients to be automatically classified into the phenotypes of erosive esophagitis, reflux hypersensitivity, functional heartburn, or nonerosive reflux disease (NERD). Additionally, it aims for patients to be automatically classified according to their manometry results using the Chicago Classification 3.0 (CC 3.0) rules and for the pH monitoring–impedance measurements of each patient to be automatically recorded after a routine examination such that the many parameters do not need to be remembered by health personnel. Therefore, this study can be used not only for scientific studies to facilitate data generation, as it is a very comprehensive database, but also for preventing errors that may arise during phenotyping. In addition, it is anticipated that the recognizability of 24 h pH impedance and/or high-resolution esophageal manometry and their classifications will increase.

## 2. Materials and Methods

This study was carried out as a pilot study to be used by the GERD study group of Ege University. In the study, as a first step, the data for 6234 patients archived between 2004 and 2017, including all of their treatment and examination data, were transferred to the developed system. Since 2017, all procedures have been performed using the developed system. Data for 2797 new patients were added to the system from 2017 to March 2020. Thus, in total, 9031 personal datasets, 5928 patient histories, 6760 endoscopy reports, 1100 classical or high-resolution manometry reports, 2462 radiology reports, 3390 consultations, 1974 reflux case discussion reports, 5609 all-drug dosage–process reports, and 4132 24 h intraesophageal impedance–pH monitoring or ambulatory capsule pH monitoring results have been included. In addition, the system has the capacity to hold data from 11 different questionnaires. These questionnaires include the Quality of Life in Reflux and Dyspepsia Questionnaire (QoLRAD) (with 12 and 25 questions), the GERD Question Forms (with 57, 66, and 81 questions), the Short Form-36 (SF-36), the Otolaryngology Form, the Otolaryngology Score, the Postop Question Form, the Reflux Disease Questionnaire (RDQ), and the Eckardt Score.

In this large and comprehensive database, patients with GERD and reflux motility problems were recorded, including details of their history, upper gastrointestinal endoscopy reports, questionnaire scores, classical or high-resolution esophageal manometry data, radiology reports, consultation reports, reflux council notes, medication doses and durations, and 24 h impedance–pH monitoring, such as their bravo capsule pH monitoring results. As a result, a decision support software package that allows examinations, questionnaires, and scores to be stored in the database, accessed upon request, decided on by the physician, and analyzed, has been created. As computer applications that make an automatic diagnosis are becoming widespread nowadays, the results of the multi-parameter pH monitoring, impedance, and symptom analysis obtained automatically over MMS (Medical Measurement Systems, The Netherlands) can be transferred to this database, as in Figure 1, and a diagnosis based on pH monitoring–impedance can be made (e.g., pH monitoring is pathologic, while impedance is normal; impedance is pathologic, while pH monitoring is normal; both of them are pathologic; pathological acid reflux; impedance is upper bounded, while pH monitoring is pathologic; etc.). As a result, the data introduced by this software via copy–paste and text parsing methods are automatically separated into 48 parameters and, thus, can save users a lot of hard work.

Figure 2 shows an example of all parameters being automatically recorded into the system after the values are entered into the system as inputs. Thus, 48 parameters can be recorded in the database in less than a minute (manually, it would take approximately 5 min). Additionally, possible input errors are prevented.

In our study, four GERD phenotypes can be automatically detected using the algorithm integrated into the system, as shown in Figure 3. Endoscopy examinations have been performed in many medical centers; therefore, the up-to-date endoscopy procedures used by the group studying reflux at Ege University are used by the developed system to determine the GERD phenotypes. If the endoscopy examination is not performed at Ege University, the most recent endoscopy examination procedure used by this other center is evaluated.

An important difference in the system is that, although many types of catheters and applications exist, the one desired can be selected and the resultant analysis can change accordingly, for example, to single-channel or dual-channel pH or bravo. Figure 4 shows the esophageal manometry page containing the embedded CC 3.0 rules; this algorithm automatically determines the phenotype of the patient. As a result, the developed system is not only an automation system but also contains rule-based algorithms on manometry, pH monitoring, and diagnosis. By means of these methods, a new original database about GERD with a large number of patients has been obtained. Based on these numbers, the Discussion section details the outcomes of this study.

Additionally, the CC 3.0 has been integrated into the system and 10 manometry diagnoses can be automatically made using the algorithm shown in Figure 5.

Another algorithmic contribution of the system is related to the questionnaires. Eleven questionnaires about GERD, adapted to Turkish, have been implemented on the system, and a digital platform through which patients can efficiently enter details from their mobile phones has been provided.

Finally, importantly, the system outputs all examinations, parameters, and results for each patient for the physicians to examine in detail. A sample of the output is given in Figure 6.

## 3. Results

Patients who underwent all examinations were evaluated in the experimental analyses. The total number of questions, total number of answers, total number of questionnaire entries, and total amount of data obtained as a result of these examinations are given in Table 1. Until March 2020, 189,765 data items were obtained with only the GERD Question Form, with 66 questions and 353 answers, while a total of 613,715 questionnaire data items were obtained for all questionnaires. This resulted in a large dataset and provided proof of the importance of studies in the field of GERD.

An advantage of the 11 questionnaires used in the system is that some data were recorded before treatment, some were recorded during treatment, and some were recorded after treatment. Thus, concealed inferences and connections can be revealed in light of the analyses performed using the common data pool that contains this big data. In addition, if these data are handled in conjunction with other examinations and treatments, hidden relations for GERD can be discovered. Additionally, such a data pool is now available for use.

Endoscopic diagnoses have the most important role in the determination of GERD phenotypes because the first step in a GERD phenotype algorithm is to check endoscopic diagnoses. For example, if a patient has an endoscopic diagnosis of esophagitis grades A, B, C, or D, a phenotype of erosive esophagitis can be determined without considering any pH monitoring results. Table 2 shows the number of each phenotype in the database, with the erosive esophagitis phenotype making up 60% of all phenotypes. Therefore, the GERD phenotype algorithm in the developed system first evaluates securable endoscopic operations, such as the operations in the reflux study group of Ege University. In the beginning, the total number of patients was 6234. However, after the data preprocessing phase, 2052 patients had features meaningful for determining their GERD phenotype. Therefore, the number of total cells in Table 2 is 2052.

Before this addition to the algorithm, the accuracy in predicting the phenotypes was approximately 90%. After the addition, the accuracy improved to 100%. These accuracy tests were performed by expert physicians, and the results were manually checked one-by-one. The diagnoses of the patients were made by gastroenterologists working in the field of GERD. Before using the AI module, the results of all patients were manually determined by these physicians according to their patient histories, endoscopic findings, classical or high-resolution manometric findings, and 24 h intraesophageal impedance–pH monitoring or ambulatory capsule pH monitoring findings, without knowing who the patients were. Then, the results of the AI module were compared with these manual results. Additionally, the confusion values, including diagnostic performance measures, were calculated. The precision, recall, and F-measure values were all 100%.

In Table 2, the second most common phenotype is non-erosive reflux disease, making up 28% of all phenotypes. Moreover, both of the most common phenotypes are 4% more likely to occur in males than in females. For the other phenotypes, the third most common phenotype is functional heartburn, making up 8% of all phenotypes, and the least common phenotype is reflux hypersensitivity, making up 4% of all phenotypes. These two phenotypes are encountered in females more often than in males. Reflux hypersensitivity is 60% more likely to occur in females than in males, and functional heartburn is 43% more likely to occur in females than in males. With respect to the age distributions in Table 2, all phenotypes are more often observed in people between 30 and 60 years old, at a rate of 71%.

For the CC 3.0 phenotypes analysis, all phenotypes in the developed database were considered, with ineffective esophageal motility making up 45% of all phenotypes. Additionally, ineffective esophageal motility is 10% more likely to occur in males than in females. Moreover, the second most common phenotype is type II achalasia, making up 23% of all phenotypes. Furthermore, type II achalasia is 35% more likely to occur in females than in males. On the other hand, EGJ outflow obstruction, distal esophageal spasm, and fragmented peristalsis phenotypes are observed in quite a few people. In addition, the other four diagnoses—type I achalasia, type III achalasia, absent contractility, and hypercontractile esophagus—are almost equally encountered in females and males. With respect to the age distributions, all phenotypes are observed more often in patients who are more than 40 years old, at a rate of 65%. Accuracy tests were performed by the physicians, and the CC 3.0 phenotype results were manually checked one by one, obtaining an accuracy of 100%. Additionally, confusion values, including the diagnostic performance measures, were calculated. The precision, recall, and F-measure values were all 100%.

Table 3 shows the numbers of each phenotype in the developed database, with the ineffective esophageal motility phenotype making up 45% of all phenotypes. Additionally, ineffective esophageal motility is 10% more likely to occur in males than in females. In Table 3, the second most common phenotype is observed to be type II achalasia, making up 23% of all phenotypes. Furthermore, type II achalasia is 35% more likely to occur in females than in males. Additionally, EGJ outflow obstruction, distal esophageal spasm, and fragmented peristalsis phenotypes are observed in quite a few people. In addition, the other four phenotypes—type I achalasia, type III achalasia, absent contractility, and hypercontractile esophagus—are nearly equally encountered in females and males. With respect to the age distributions in Table 3, all phenotypes are observed more often in patients more than 40 years old, at a rate of 65%. Accuracy tests for the CC 3.0 phenotypes were again manually performed by the physicians, and the results were checked one by one, obtaining an accuracy of 100%.

Figure 7 shows the distribution of the number of patients from January 2015 to March 2020. The developed system started to store patient data in the database in 2017. The number of patients included in 2015, 2016, 2017, 2018, and 2019 was 373, 372, 577, 794, and 652, respectively. The number of patients included in January and February 2020 was 129; so, for all months in 2020, we predicted that 129 × 6 = 774 patients could be stored in the database. Thus, it is noted that, since a transition to using the developed system in 2017, it has been observed that the number of patients, which was around 400 beforehand, increased to 800.

Moreover, the database is accessible only on campus, with any off-campus access only available with explicit permission from the IT Department of Ege University. Data security was ensured by following personal data protection procedures. Finally, it can be noted that this automation and decision support system improves the performance of physicians by nearly two times for patient care, diagnosis, and treatment management.

## 4. Discussion

In the field of gastroenterology, a medical information system was first implemented in 1984. That system had a DSS with a simple knowledge base and statistical structure [12]. Reporting by querying databases was seen next, as described in another gastroenterology study [13], and examples developed for specific purposes, such as drug tracking systems, were then implemented [14]. Currently, AI studies in gastroenterology have raised awareness about this subject worldwide [15,16]. These studies were performed to search for solutions for different sub-disciplines in gastroenterology; however, no comprehensive information system relating to GERD has so far been widely used in the world. Additionally, the main epidemiology studies were conducted in developed western countries. However, Turkiye has a different GERD profile. While the main complaint presented in developed western countries is a burning sensation behind the breastbone, in Turkiye, regurgitation is the most common symptom. Similarly, the Barrett problem, which is a cancer-related subgroup of reflux, has a prevalence of 10% in developed countries, while its prevalence is about 1% in Turkiye. Moreover, erosive esophagitis C and D are also less common [17,18]. Therefore, to represent the different realities in Turkiye, storing data in an environment where comprehensive analyses can be carried out will be of value. However, a central database of all records and data files stored in Turkiye has not previously been available until now.

This study represents the first time that a database and information system for GERD in Turkiye has been developed and published to improve medical workflow, to monitor patients, and to help physicians make decisions at various stages using machine learning algorithms.

Studies about reflux previously performed by our group have been highly cited and have come to the fore in the literature [19]. These studies could only have been carried out with a large number of patients. Additionally, our studies have been referenced in recent publications [17]. In order to achieve this structure, which includes an increasing number of patients, extensive computer support has been required. All the requirements for the automation and decision support system were influenced by feedback from our team.

Machine learning and deep learning techniques, which have demonstrated significant benefits and shown successful results, are also used in the field of gastroenterology. Illustrative examples include the following: automatic endoscopic scoring was performed using machine learning for ulcerative colitis, which manifests over the long term [17]; a machine-learning-based scoring system was developed to screen for high-risk esophageal varicose veins [20]; a machine learning model with better performance than clinical risk scoring systems for upper gastrointestinal bleeding was implemented [21]; machine learning algorithms were used to classify patients with constipation [22]; a deep learning model that can detect anterior gastric cancer symptoms was developed [23]; and, using another deep learning model, endoscopic diagnosis and treatment planning were implemented for colorectal polyps [24]. As a result of using this system in the successful studies described, a large database has been created and the use of machine learning and deep learning techniques has been facilitated.

Patients’ medicinal treatment and responses to proton pump inhibitors (PPI) have been recorded in the developed system, which has allowed for quality-of-life and PPI response studies to be conducted. For example, up-to-date PPI threshold values specific to GERD phenotypes can be determined without fixing the PPI response to 50% [25]. Furthermore, the validity and reliability of the QoLRAD questionnaire in patients with gastroesophageal reflux disease for the Turkish population have been assessed [26].

Examples of studies implemented to identify sub-phenotypes, such as studies using automated phenotyping for type 2 diabetes [27], as well as studies determining the sub-phenotype of liver diseases using hierarchical clustering [28], were identified. By means of the developed system, GERD patients are automatically clustered into the phenotypes of erosive esophagitis, reflux hypersensitivity, functional heartburn, or NERD. Moreover, GERD patients can be automatically classified according to their manometry results using the CC 3.0 rules specified in [29].

In conclusion, healthcare personnel can now access information from any location using a mobile device, such as a cell phone or tablet, and due to the capabilities of the developed system, health personnel’s efficiency in caring for patients has increased. Additionally, AI studies on reflux have increased [30,31,32].

## Figures and Tables

**Figure 1 healthcare-11-01790-f001:**
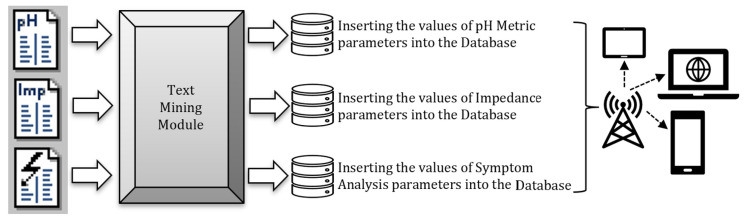
The process of mining “pH monitoring—impedance—symptom analysis” reports, storing their parameters in a database, and publishing them for access on mobile communication devices, such as tablets, laptops, cell phones, etc.

**Figure 2 healthcare-11-01790-f002:**
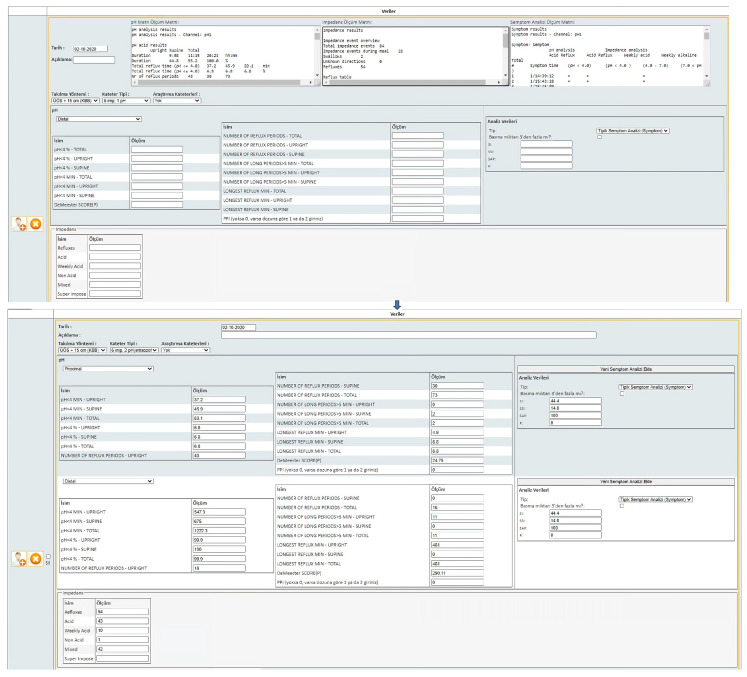
The “pH Monitoring–Impedance–Symptom Analysis” page.

**Figure 3 healthcare-11-01790-f003:**
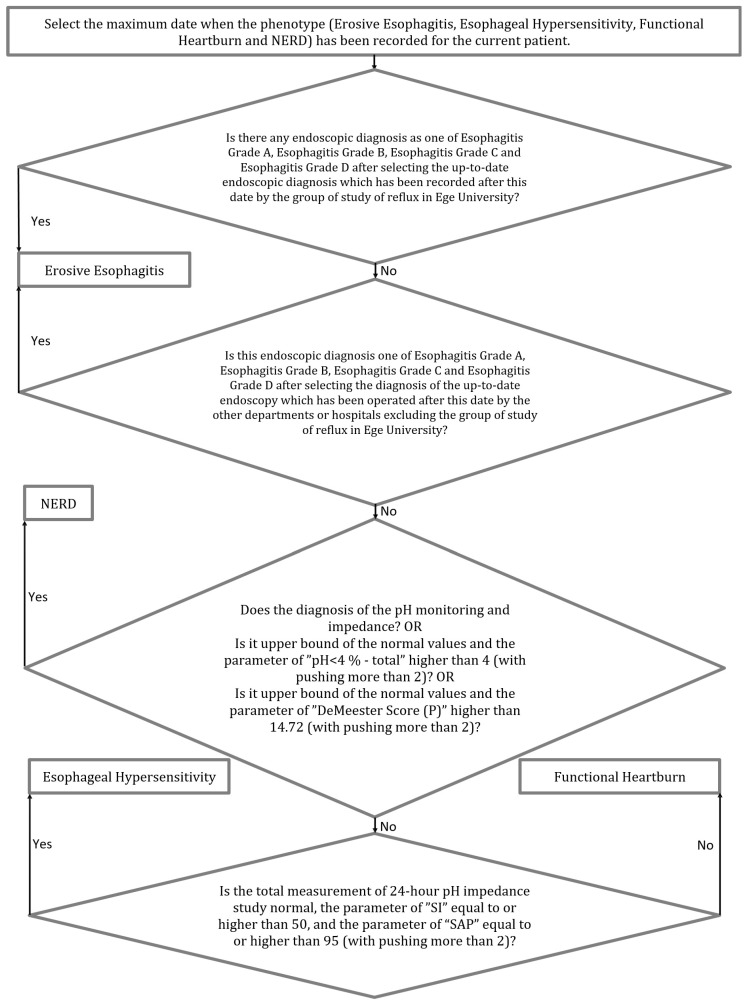
Flowchart of the GERD phenotype rules.

**Figure 4 healthcare-11-01790-f004:**
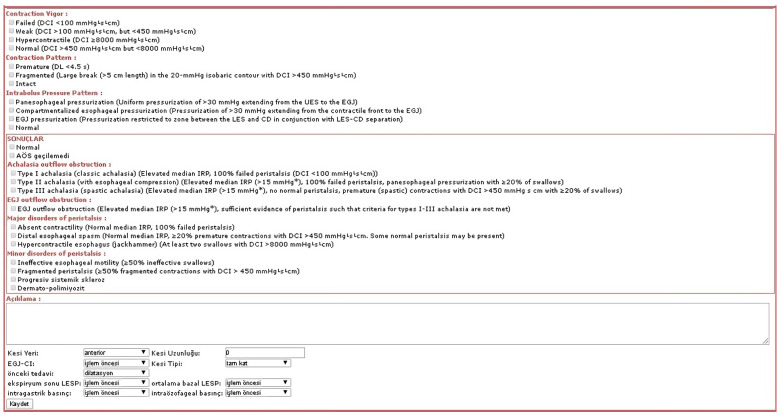
The “Manometry Analysis” page.

**Figure 5 healthcare-11-01790-f005:**
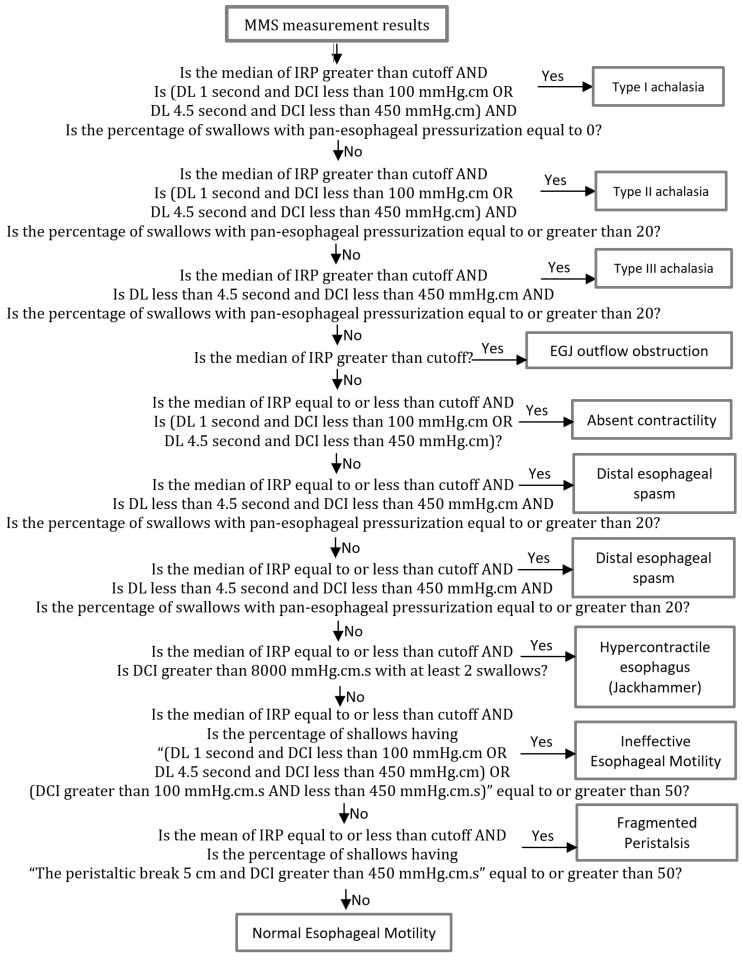
Flowchart of the CC 3.0 rules.

**Figure 6 healthcare-11-01790-f006:**
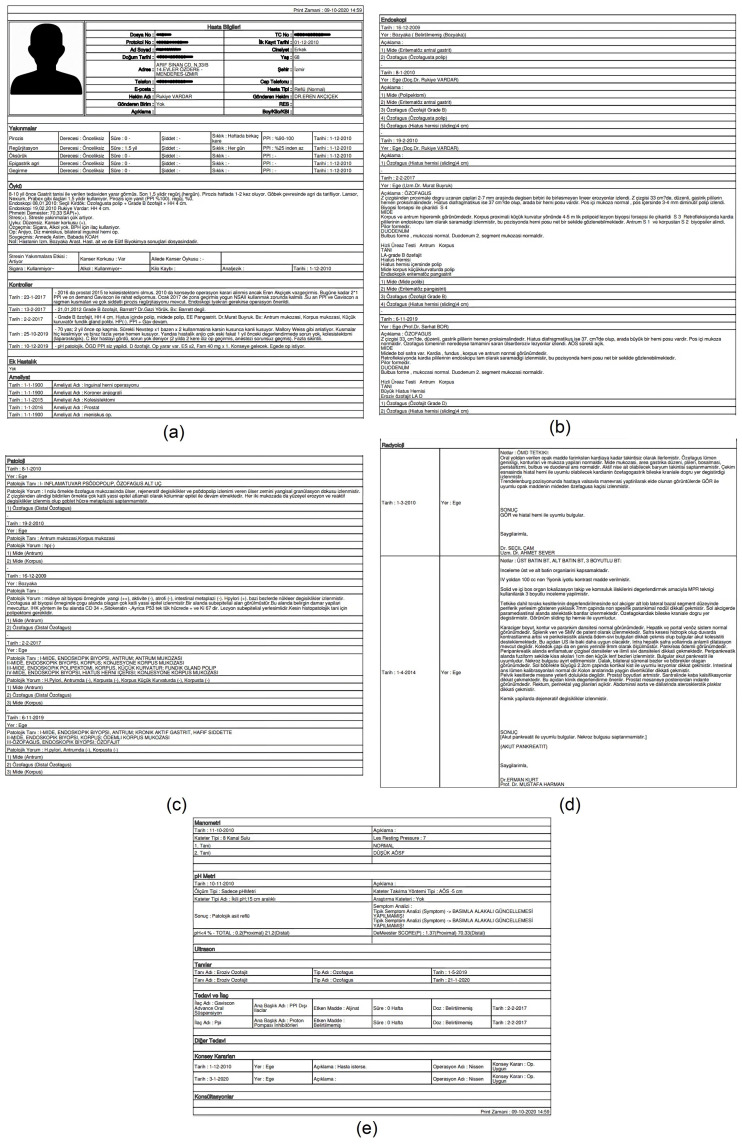
A general overview of the examinations, the parameters, and their results for a sample patient: (**a**) history, complaints, controls, additional diseases, and operations; (**b**) endoscopic values; (**c**) pathologic values; (**d**) radiologic values; (**e**) manometric values, pH monitoring, ultrasound results, diagnosis, medicine, other treatments, council decisions, and consultations.

**Figure 7 healthcare-11-01790-f007:**
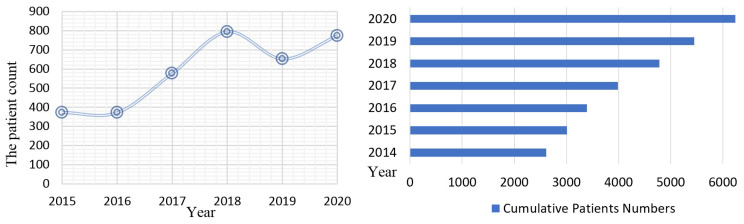
A graph of the patient count distributions from 2015 to 2020.

**Table 1 healthcare-11-01790-t001:** Questionnaires and their total numbers of questions, answers, and entries.

	Total Questions	Total Answers	Total Entries	Total Amount of Data
QoLRAD1	12	84	4276	48,917
QoLRAD2	25	175	1723	38,800
GERD Question Form 1	57	238	653	33,942
GERD Question Form 2	66	353	5041	189,765
GERD Question Form 3	81	444	1873	185,774
SF-36	11	149	5399	119,252
Otolaryngology Form (11)	20	115	1446	21,196
Otolaryngology Score (11)	9	28	1602	10,603
GERD Postoperative Symptoms Question Form	22	96	156	2922
RDQ	2	72	82	906
Eckardt Score	5	17	10	50
Total	310	1771	22,261	613,715

**Table 2 healthcare-11-01790-t002:** Sociodemographic characteristics of the GERD phenotypes.

	Erosive Esophagitis (EE)	Reflux Hypersensitivity (RH)	Functional Heartburn (FH)	Non-Erosive Reflux Disease (NR)	Total
Male	641	12	48	307	1008
Female	590	49	121	284	1044
Age (10–19)	18	4	2	10	34
Age (20–29)	119	8	18	60	205
Age (30–39)	271	19	42	127	459
Age (40–49)	298	13	49	149	509
Age (50–59)	299	13	37	144	493
Age (60–69)	169	3	19	77	268
Age (70–90)	57	1	2	24	84
Total	1231 (60%)	61 (3%)	169 (8%)	591 (29%)	2052

**Table 3 healthcare-11-01790-t003:** Socio-demographic characteristics for the CC 3.0 phenotypes.

	Male	Female	Age < 40	Age ≥ 40	Total
Type I achalasia (classic achalasia)	7	7	5	9	14 (11%)
Type II achalasia (with esophageal compression)	10	21	10	21	31 (23%)
Type III achalasia (spastic achalasia)	4	4	1	7	8 (6%)
EGJ outflow obstruction	1	0	0	1	1 (1%)
Absent contractility	4	4	2	6	8 (6%)
Distal esophageal spasm	1	0	0	1	1 (1%)
Hypercontractile esophagus (jackhammer)	5	4	0	9	9 (7%)
Ineffective esophageal motility	37	23	29	31	60 (45%)
Fragmented peristalsis	1	0	0	1	1 (1%)
Total	70	63	47	86	133

## Data Availability

The data are available upon request via correspondence. Ethical committee confirmation number: 2017-5.1/49.

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
