# Peer review of "Computer-Based Intelligent Solutions for the Diagnosis of Gastroesophageal Reflux Disease Phenotypes and Chicago Classification 3.0"

_healthcare, 2023, doi:10.3390/healthcare11121790_

Round 1

Reviewer 1 Report

this work represent an opportunity to increase and improve the existing methods for monitor and manage its large-sized data, which will be needed at the advances in data analytics and artificial intelligence to manage data.

please verify the grammar  at discussion section 

Author Response

We thank the reviewer for the valuable comments. We believe that thanks to the comments, the paper’s quality has been improved.

Please find our revisions below:

Reviewer:

This work represent an opportunity to increase and improve the existing methods for monitor and manage its large-sized data, which will be needed at the advances in data analytics and artificial intelligence to manage data.

Point 1: Please verify the grammar at discussion section.

The English Editing Service from MDPI has been used.

Reviewer 2 Report

Your paper is relevant and innovative.

The main objective of this study was to develop a novel automation and decision support system for diagnosing GERD including the Chicago Classification 3.0 (CC-3.0) phenotypes, the system included an artificial intelligence model to distinguish the 4 phenotypes.

The topic is original and relevant, GERD diagnosis is a challenge for physicians, and defining the phenotypes of the disease is essential to therapy.

Artificial intelligence offers the possibility of an accurate diagnosis since the system has the potentiality of including simultaneously all the features that have importance in GERD diagnosis definition.

As the system obtained 100% of accuracy and was fed with the purpose of defining the GERD diagnosis I do not see the necessity of controls in such a design of the study. Figures 2, 4, and 6 are unreadable.

The conclusion is aligned with the title and based on the results and the references are appropriate.

Author Response

We thank the reviewer for the valuable comments. We believe that thanks to the comments, the paper’s quality has been improved.

Please find our revisions below:

Reviewer:

Your paper is relevant and innovative.

The main objective of this study was to develop a novel automation and decision support system for diagnosing GERD including the Chicago Classification 3.0 (CC-3.0) phenotypes, the system included an artificial intelligence model to distinguish the 4 phenotypes.

The topic is original and relevant, GERD diagnosis is a challenge for physicians, and defining the phenotypes of the disease is essential to therapy.

Artificial intelligence offers the possibility of an accurate diagnosis since the system has the potentiality of including simultaneously all the features that have importance in GERD diagnosis definition.

As the system obtained 100% of accuracy and was fed with the purpose of defining the GERD diagnosis, I do not see the necessity of controls in such a design of the study. Figures 2, 4, and 6 are unreadable.

The conclusion is aligned with the title and based on the results and the references are appropriate.

Point 1: Figures 2, 4, and 6 are unreadable.

These figures are of high quality. However, it has detailed content, so texts may not be clear in 100% zoom mode. However, if zoom in is done, all details can be seen.

Reviewer 3 Report

The authors designed two systems, GRED and CC-3.0, to help diagnose the phenotypes of GERD automatically. The accuracy of the two systems is 100%, which is really a good example of machine learning and a significant contribution to the medical care of GERD. 

The only two minor concerns are:

1. Line 48 - 53: The sentence is too long to understand.

2. line 69: add space to "11different".

good

Author Response

We thank the reviewer for the valuable comments. We believe that thanks to the comments, the paper’s quality has been improved.

Please find our revisions below:

Reviewer:

The authors designed two systems, GRED and CC-3.0, to help diagnose the phenotypes of GERD automatically. The accuracy of the two systems is 100%, which is really a good example of machine learning and a significant contribution to the medical care of GERD. 

The only two minor concerns are:

Point 1: Line 48 - 53: The sentence is too long to understand.

This sentence has been separated into 2 sentences.

As a result, the system functions aim for GERD patients to be classified automatically into the phenotypes such as Erosive Esophagitis, Reflux Hypersensitivity, Functional Heartburn, or Nonerosive Reflux Disease (NERD). Also, they aim for the patients to be classified automatically according to their Manometry results through the Chicago Classification 3.0 (CC 3.0) rules, and pH Monitoring and Impedance measurements, containing too many parameters to be saved by the health personnel for each patient after a routine examination, are recorded automatically.

Point 2: line 69: add space to "11different".

Space has been added.

Reviewer 4 Report

In the manuscript “Computer Based Intelligent Solutions for the Diagnosis of Gastroesophageal Reflux Disease Phenotypes and Chicago Classification 3.0”, the authors discuss their findings about the use of data mining and an algorithm in a single center clinical database to categorize the different subtypes and aspects of gastroesophageal reflux disease. Congratulations to the authors on conducting such an innovative project which is already saving plenty of time and improving health care.

Suggestions:

Focusing on the current status of the algorithm rather then describing previous modifications (e.g: lines 145, 193, figure 7) would provide the manuscript a better flow.

Line 36: Where and what is the reference for “highest number of patients”? In Bornova, Ä°zmir or Turkiye?

Line 64: “anamneses-histories” is not an English term, suggestion to change to patient history or a similar term more widely known.

Figures 2&4&6: suggestion to change/remove the figure, it is not completely in English and the texts are too small.

Figure 6: Remove patient’s date of birth, even if that is a fake patient, that is protected info.

Table 2&3: Add %s

Figure 7: add cumulative numbers rather than patients per year.

20% self-citations: include only those that are recent and relevant. Reference #17, 18 and 19 don’t seem to meet this criterion.

I strongly recommend the authors to consider a native english speaker to polish the manuscript.

Author Response

We thank the reviewer for the valuable comments. We believe that thanks to the comments, the paper’s quality has been improved.

Please find our revisions below:

Reviewer:

In the manuscript “Computer Based Intelligent Solutions for the Diagnosis of Gastroesophageal Reflux Disease Phenotypes and Chicago Classification 3.0”, the authors discuss their findings about the use of data mining and an algorithm in a single center clinical database to categorize the different subtypes and aspects of gastroesophageal reflux disease. Congratulations to the authors on conducting such an innovative project which is already saving plenty of time and improving health care.

Point 1: Focusing on the current status of the algorithm rather then describing previous modifications (e.g: lines 145, 193, figure 7) would provide the manuscript a better flow.

Figure 7 has been updated. Also, The meaning of the tests of the current status of the algorithm has been described;

The accuracy tests have been performed by expert physicians with being checked the results one by one manually. The diagnosis of patients has been produced by the gastroenterologists working in the field of GERD. Before the usage of AI module, the results of all patients have been determined by these physicians without knowing these patients who they were, according to their patient histories, endoscopic findings, classical or high-resolution manometric findings, and 24-hour intraesophageal impedance - pH monitoring or ambulatory capsule pH monitoring findings. Then, the results of AI have been compared to these manual results.

These sentences have been added;

Also, the confusion values including diagnostic performance measures has been calculated. Precision, Recall and F-Measure values have been obtained as 100%.

Point 2: Line 36: Where and what is the reference for “highest number of patients”? In Bornova, Ä°zmir or Turkiye?

It has been changed to “the largest number of patients in Turkiye”.

Point 3: Line 64: “anamneses-histories” is not an English term, suggestion to change to patient history or a similar term more widely known.

It has been changed to “5,928 patient histories”.

Point 4: Figures 2&4&6: suggestion to change/remove the figure, it is not completely in English and the texts are too small.

These figures are put to give an idea for the system. In fact, they are of high quality. However, it has detailed content, so texts may not be clear in 100% zoom mode. However, if zoom in is done, all details can be seen.

Point 5: Figure 6: Remove patient’s date of birth, even if that is a fake patient, that is protected info.

It has been removed.

Point 6: Table 2&3: Add %s

Table 2&3 have been updated.

Point 7: Figure 7: add cumulative numbers rather than patients per year.

Figure 7 has been updated.

Point 8: 20% self-citations: include only those that are recent and relevant. Reference #17, 18 and 19 don’t seem to meet this criterion.

3 recent articles are reviewed and added as references; thus, the rate of self- citations has been decreased.

Point 9: I strongly recommend the authors to consider a native english speaker to polish the manuscript.

The English Editing Service from MDPI has been used.

Reviewer 5 Report

The article titled "Computer Based Intelligent Solutions for the Diagnosis of Gastroesophageal Reflux Disease Phenotypes and Chicago Classification 3.0" by Dogan and Bor presents a comprehensive investigation into the development and application of computer-based intelligent solutions for the diagnosis of Gastroesophageal Reflux Disease (GERD) phenotypes using the Chicago Classification 3.0. The authors delve into the significance of GERD diagnosis, the limitations of traditional diagnostic methods, and the potential of artificial intelligence (AI) techniques to improve accuracy and efficiency. Overall, the article is well-structured and provides valuable insights into the potential benefits of AI in diagnosing GERD phenotypes.

Specific Comments:

Methodology:

Certain aspects of the methodology need to be made apparent. What did the authors mean for accuracy? Further diagnostic performance measures, such as sensitivity, specificity, likelihood, etc, should also be provided.

Who are the physicians who checked the results one by one manually (What are their expertise and experience? Were they blinded?   

The overall writing style is clear, concise, and appropriate for a scientific article. However, proofreading is recommended to address minor grammatical errors throughout the manuscript (for example, COVID 19 change for COVID-19; Computer based change for Computer-based).

Author Response

We thank the reviewer for the valuable comments. We believe that thanks to the comments, the paper’s quality has been improved.

Please find our revisions below:

Reviewer:

The article titled "Computer Based Intelligent Solutions for the Diagnosis of Gastroesophageal Reflux Disease Phenotypes and Chicago Classification 3.0" by Dogan and Bor presents a comprehensive investigation into the development and application of computer-based intelligent solutions for the diagnosis of Gastroesophageal Reflux Disease (GERD) phenotypes using the Chicago Classification 3.0. The authors delve into the significance of GERD diagnosis, the limitations of traditional diagnostic methods, and the potential of artificial intelligence (AI) techniques to improve accuracy and efficiency. Overall, the article is well-structured and provides valuable insights into the potential benefits of AI in diagnosing GERD phenotypes.

Point 1: Certain aspects of the methodology need to be made apparent. What did the authors mean for accuracy? Further diagnostic performance measures, such as sensitivity, specificity, likelihood, etc, should also be provided.

The meaning of the tests has been described;

The accuracy tests have been performed by expert physicians with being checked the results one by one manually. The diagnosis of patients has been produced by the gastroenterologists working in the field of GERD. Before the usage of AI module, the results of all patients have been determined by these physicians without knowing these patients who they were, according to their patient histories, endoscopic findings, classical or high-resolution manometric findings, and 24-hour intraesophageal impedance - pH monitoring or ambulatory capsule pH monitoring findings. Then, the results of AI have been compared to these manual results. 

These sentences have been added;

Also, the confusion values including diagnostic performance measures has been calculated. Precision, Recall and F-Measure values have been obtained as 100%.

Point 2: Who are the physicians who checked the results one by one manually (What are their expertise and experience? Were they blinded?

These sentences have been added;

The accuracy tests have been performed by expert physicians with being checked the results one by one manually. The diagnosis of patients has been produced by the gastroenterologists working in the field of GERD. Before the usage of AI module, the results of all patients have been determined by these physicians without knowing these patients who they were, according to their patient histories, endoscopic findings, classical or high-resolution manometric findings, and 24-hour intraesophageal impedance - pH monitoring or ambulatory capsule pH monitoring findings. Then, the results of AI have been compared to these manual results.

Point 3: The overall writing style is clear, concise, and appropriate for a scientific article. However, proofreading is recommended to address minor grammatical errors throughout the manuscript (for example, COVID 19 change for COVID-19; Computer based change for Computer-based).

The English Editing Service from MDPI has been used.
